# Table Learning Representation from Scanned PDF Documents Containing Some Red Stamps

## Abstract

Generally, it can be challenging to recognize the table contents from scanned PDF documents containing some red stamps and reconcile the recognized table contents. In this paper, we address the reconciliation challenge involving matching the handwritten invoice amounts overlapped with a red stamp against the invoice amounts found in multi-page tables extracted from scanned PDF documents and we propose a context-splitting Transformer-based table recognition method for recognizing the handwritten invoice amounts overlapped with a red stamp against the invoice amounts. Firstly, we recognize the layout structure of the table which is detected from the scanned PDF document containing some red stamps on the handwritten invoice amounts of the table. Secondly, we represent the table cells as context-splitting embedding vectors which involve spatial context embedding, position context embedding, lexical context embedding, and colored context embedding. Finally, we apply a stack of Transformer-based self-attention encoders to recognize the cross-modality table cells where we multiply the length of query vector and the length of key vector with the scaling factor of the original Transformer in order to make the training process more stable. We improve the recognition accuracy of table cells with a red stamp on handwritten invoice amounts.

## 1 Introduction

Tables are a common form in Portable Document Format (PDF) documents. Especially, some tables in scanned PDF documents contain some handwritten signatures and some red stamps on them. We present an issue in scanned PDF recognition, namely red stamp occlusions and the recognition of handwritten text under the red stamp. However, it can be challenging to recognize such table contents as handwritten invoice amounts from scanned PDF documents containing some red stamps and reconcile the recognized table contents. For example, PP-StructureV2 Li et al. (2022a) which contains a subsystem "Layout Information Extraction" and subsystem "Key Information Extraction" can detect the table structures and recognize the table cells from the scanned PDF documents. However, the effectiveness will be greatly reduced when a PDF document is obtained by scanning the printed document and the tables in this PDF document involve some handwritten invoice amounts and some red stamps on these handwritten invoice amounts. Figure 1 shows an example of the results on table structure detection and table cell recognition from the scanned PDF documents which involve some handwritten invoice amounts and some red stamps on them by importing PPStructure and applying it. In Figure 1, PPStructure divides the document image of left side into predefined areas such as text, title, table, and figure based on an ultra lightweight detector PP-PicoDet. For example, the table symbol on the upper left corner in Figure 1 represents a table area identified by PP-PicoDet detector. However, the table cell recognition performance need to be improved in the invoice amount column containing a red stamp on a handwritten invoice amount, shown as the red arrow in Figure 1. There are at least three errors on the recognition results of table elements under the red stamp in Figure 1. Firstly, the handwritten invoice amount with a red stamp on it was completely unrecognized and its recognition result was empty in the column of "Invoice amount" (in Chinese), shown as the blue box in the left side of Figure 1. Secondly, the invoice number "13645010" was incorrectly identified as "3645010" in the column of "Invoice number" (in Chinese), shown as the green box in the left side of Figure 1. Thirdly, the amount "420929.72" was incorrectly recognized as an amount "9.9420929.7" in the column of "Transfer price of accounts receivable debt" (in Chinese), shown as the yellow box in the left side of Figure 1.

Figure 1: The result on table structure detection and table cell recognition from the scanned PDF documents containing red stamps on some handwritten invoice amounts using PPStructure.

In order to correctly recognize the handwritten invoice amounts under a red stamp in the column "invoice amount" (in Chinese) of the image-based table in Figure 1, we propose a color-wise context-splitting Transformer-based table recognition method for recognizing the contents of table cells and the handwritten invoice amounts overlapped with a red stamp against the invoice amounts. The contributions of our proposed color-wise context-splitting Transformer-based method for recognizing the table cells are three-folds.

- Firstly, we recognize the layout structure of the table which is detected from the scanned PDF document with some red stamps or stamps on some handwritten invoice amounts of the table.
- Secondly, we represent the table cells as context-splitting embedding vectors which involve spatial context embedding, position context embedding, lexical context embedding, and colored context embedding.
- Finally, we apply a stack of Transformer-based self-attention encoders to recognize the cross-modality table cells where we multiply the length of query vector and the length of key vector with the scaling factor of the original Transformer in order to make the training process more stable.

As a result, we improve the recognition accuracy of table cells with a red stamp or red stamp on handwritten invoice amounts and the efficiency of reconciliation.

## 2 RELATED WORK

### 2.1 TABLE DETECTION FROM SCANNED PDF DOCUMENTS

Table detection from scanned PDF documents is a necessary step for localizing the bounding boxes of tables in PDF documents Zhong et al. (2020). TableSense Dong et al. (2019) is a spreadsheet Table Detection framework with Convolutional Neural Network (CNN). Multi-Type-TD-TSR Fischer et al. (2021) uses a fully data-driven approach based on a CNN. Graph Neural Network (GNN) is used to describe the local repetitive structural information of tables in invoice documents Riba et al. (2021). PSENet can be used to detect each text line in the table image Ye et al. (2021). CatBoost Prokhorenkova et al. (2018) is the SOTA Tree-based boosting model on tabular tasks. TableFormer Nassar et al. (2022) provides bounding boxes by using Cell BBox Detector which can be improved with post-processing during inference. However, both PSENet and TableFormer are not very ideal to detect curve texts such as red stamps or stamps. Differentiable Binarization Liao et al. (2020) Liao et al. (2023) can perform the binarization process in a segmentation network which can more accurately describe scene text of various shapes such as curve text. PP-OCR Du et al. (2020) uses Differentiable Binarization as text detector which is based on a simple segmentation network. The simple post-processing of Differentiable Binarization is efficient. PP-OCRv2 Du et al. (2021) uses Collaborative Mutual Learning (CML) as text detector to solve the problem of text detection distillation. In CML method, text detector uses ResNet18 as teacher model and MobileNetV3 as student model. PP-OCRv3 Li et al. (2022b) still uses CML as text detector which adopts a Large Kernel PAN (LK-PAN) module for a teacher model and a RSE-FPN module with residual attention mechanism for a student model. PP-StructureV2 Li et al. (2022a) introduces a lightweight layout detector PP-PicoDet to divide document images into predefined areas such as text, title, table, and figure.

## 2.2 TABLE RECOGNITION OF SCANNED PDF DOCUMENTS

Table recognition of scanned PDF documents is an important step for parsing only the structural information on row and column layout of tables Zhong et al. (2020) and extracting the logical and physical structure of unstructured table images into a machine-readable format Lin et al. (2022)Huang et al. (2023). Table Structure Recognition (TSR) involves bordered TSR, Unbordered TSR, and Partially Bordered TSR Fischer et al. (2021). Cycle-CenterNet Long et al. (2023) on the top of CenterNet detects and groups tabular cells into structured tables. A bi-directional Recurrent Neural Network (RNN) with Gated Recurrent Units (GRU) Khan et al. (2020) is used to extract rows and columns from a detected table in document images. Cycle-CenterNet Long et al. (2021) detects and groups tabular cells into structured tables. TGRNet Xue et al. (2021) reformulates the problem of table structure recognition as the table graph reconstruction. Donut Kim et al. (2022) is an OCR-free document Understanding Transformer which is robust to the handwritten documents. TabTransformer Huang et al. (2020) architecture comprises a column embedding layer to learn efficient contextual of table, a stack of N Transformer layers, and a Multi-layer Perceptron (MLP). FT-Transformer Gorishniy et al. (2021) transforms all categorical and numerical features of tabular data to embeddings by Feature Tokenizer and applies a stack of Transformer layers to the embeddings. MATE Eisenschlos et al. (2021) presents a sparse-attention Transformer architecture for modeling documents that contain large tables. TRUST guo et al. (2022) is an end-to-end Transformer-based table structure recognition method with multi-oriented table row/column splitting step and table grid merging step. DETR Carion et al. (2020) employs a Transformer Encoder and Decoder that looks for a specific number of object queries. TableFormer Nassar et al. (2022) replaces the LSTM decoders with Transformer-based decoders which can obtain the content of the table-cells from PDF source with some non-English tables by using bounding boxes. TSRFormer Lin et al. (2022) contains a two-stage DETR-based split module to predict all row/column separation lines from each input table image and a relation network based cell merging module to recover spanning cells. VAST Huang et al. (2023) contains a CNN image encoder, a HTML sequence decoder which is a Transformer with a stack of N identical layers, a coordinate sequence decoder triggered by the representation of the non-empty cell from HTML sequence decoder where the logical structure of the table is represented in HTML format. XTab Zhu et al. (2023) handles the cross-table pre-training of tabular Transformers on datasets from various domains by separating the models into data-specific and shared components to learn general knowledge for tabular prediction. LayoutLMv2 Xu et al. (2021) is a multi-modal pre-training method for visually-rich document understanding. PP-StructureV2 Li et al. (2022a) use a visual-feature independent LayoutXLM architecture to extract key information.

## 3 METHODOLOGY

In order to correctly recognize the handwritten invoice amounts under a red stamp in the column "invoice amount" (in Chinese) of the image-based table in Figure 1, we propose a color-wise context-splitting Transformer-based method for analyzing the table structures and recognizing the table cells with red stamp occlusions and handwritten invoice amount. Figure 2 shows an architecture of our proposed color-wise context-splitting Transformer-based method for recognizing the table cells with red stamp occlusions and handwritten invoice amount. In Figure 2, our proposed color-wise context-splitting Transformer-based method includes at least six parts for analyzing the table structures and recognizing the table cells with red stamp occlusions and handwritten invoice amount. The first part is an input layer which inputs the table image containing a red stamp in the table detected from a multi-pages scanned PDF document. The second part is a table image standardization layer where the direction of input table image is corrected through a table alignment pre-processing step. The third part is a table layout structure analysis layer where the physical structure of a table image refers to the bounding box coordinates of all non-empty table cells and the logical structure of this table is its layout structure sequence. The forth part is an embedding layer where we represent the spatial contexts, positional contexts, lexical (tokenized) contexts, and colored (tinctorial) contexts of the table cells in a layout structure sequence as low dimensional embedding vectors. The fifth part is a stack of Transformer-based layers where we modify the self-attention mechanism of the original Transformer Vaswani et al. (2017) by multiplying the length of query vector and the length of key vector with the scaling factor of the original Transformer in order to make the training process more stable. The sixth part is an output layer which outputs the probabilities of predicted table cells in a table layout structure sequence.

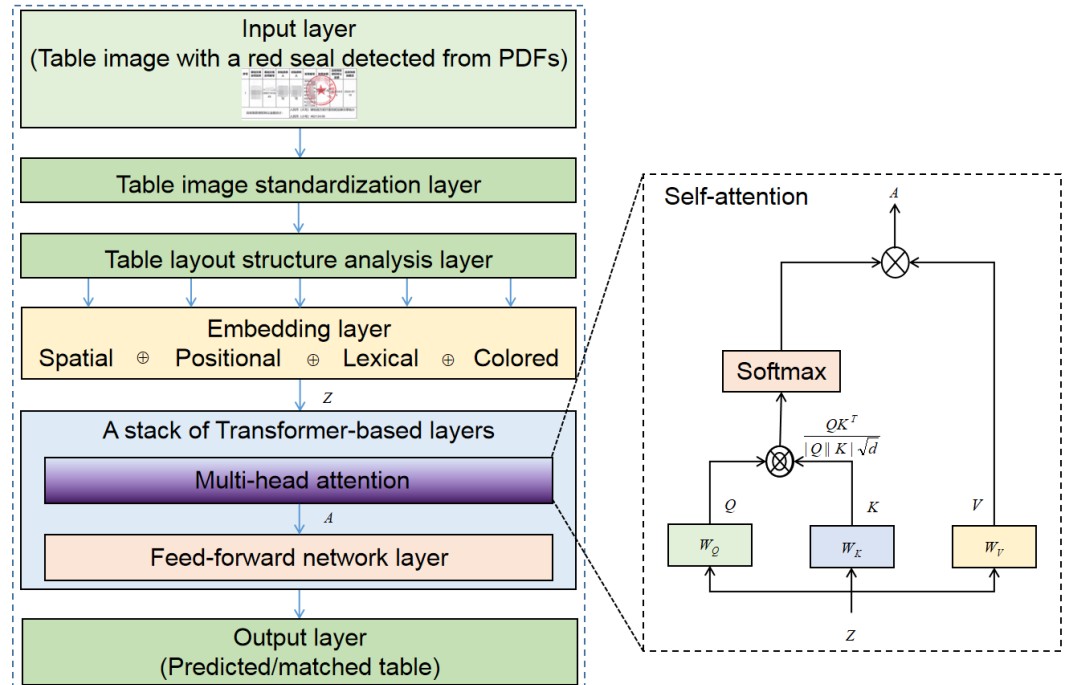

Figure 2: An architecture of our proposed color-wise context-splitting Transformer-based method for recognizing the table cells with red stamp occlusions and handwritten invoice amount.

## 3.1 TABLE LAYOUT STRUCTURE ANALYSIS

The purpose of the table layout structure analysis is to reconstruct the HTML sequence items with the sequential relationship cross table cells and their corresponding locations on the table but ignore the text contents in each item. The HTML representation of a table structure can be easily converted into Excel files. For example, a PPStructure Li et al. (2022a) backbone is employed to convert a table image into a HTML sequence. In PPStructure backbone, PP-PicoDet uses CSP-PAN (Cross Stage Partial Path Aggregation network) as a neck. Compared to CSP, XTab Zhu et al. (2023) can leverage the information across multiple tables in PDF documents. So, we adopt XTab-PAN as a neck to analyze the table layout structure. Figure 3 shows an architecture of our proposed XTab-PAN based detector and context-splitting based decoder for analyzing the table layout structure. In Figure 3, the table image detected from the scanned PDF document is inputted into the XTab-

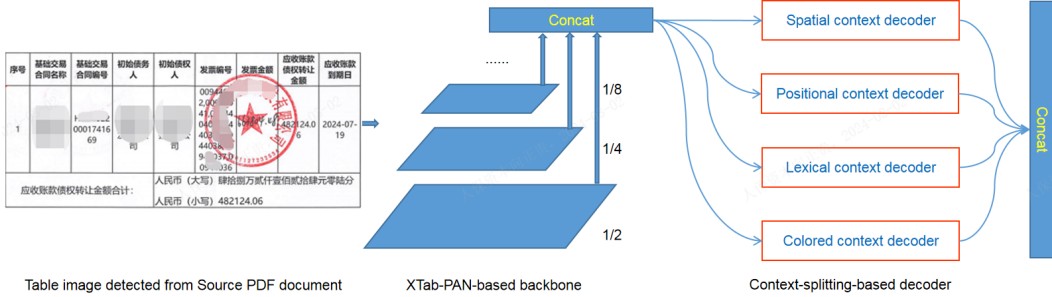

Figure 3: An architecture of our proposed XTab-PAN based detector and context-splitting based decoder for analyzing the table layout structure.

PAN based detector for feature extraction. The XTab-PAN based backbone network detector obtains the feature maps of different sizes such as 1/2 and 1/4 which detect small and large targets in the table image separately, and then fuses these features with different sizes through a feature fusion

network (Neck) which sends the fused features to the context-splitting based decoder. Finally, we construct efficiently the feature representation of each table cell by fusing spatial, positional, lexical, and colored context features extracted from the context-splitting based decoders. For example, $(tablecell_1\{row_1, col_1, order_1, token_1, red_1, green_1, blue_1\}, boundingbox_1\{x_1, y_1, w_1, h_1\})$ indicates a HTML sequence item of a table layout structure where $tablecell_1$ is a table cell in the table layout structure with its {row,column} spatial context $\{row_1, col_1\}$, positional context $\{order_1\}$, lexical context $\{token_1\}$, and colored context $\{red_1, green_1, blue_1\}$ and $boundingbox_1$ is the bounding box of the table cell with a {horizontal,vertical,width,height} coordinate $\{x_1, y_1, w_1, h_1\}$.

## 3.2 EMBEDDING REPRESENTATION

Firstly, we represent the spatial contexts of the table cells in the table layout structures as low dimensional embedding vectors. Let $ES_k, k \in [1, N]$ mean the embedding vector with dimension $d$ of the $k^{th}$ table cell, where $N$ means the number of all table cells $(d < N)$. We note a cross-entropy loss function of spatial context embedding as $L_{ES}$ and split $ES_k$ as (row,col) embedding $(ES_{k,row}, ES_{k,col})$ shown as Equation (1).

$$ES_k = ES_{k,row} + ES_{k,col} \tag{1}$$

Secondly, we represent the positional contexts of the table layout structures in the table as low dimensional embedding vectors. A complex-valued function is a potential continuous function to capture the ordinal relationship between positions. In original Transformer Vaswani et al. (2017), each dimension of the position embedding is obtained based on the discrete position indexes, which makes it difficult to model the ordinal or ordered relationship between positions such as adjacency or precedence. To fully capture the positional aspect, one solution is to build complex-valued continuous function over the position index to capture smooth transformation from a position to its adjacent position Wang et al. (2020). The representation in different position can correlate with each other in complex-valued continuous function. Therefore, we use a complex-valued continuous function to represent the sparse ordinal relationship in the layout structures. Let $EP_k$ mean the embedding vector with dimension $d$ of the ordinal relationship for the $k^{th}$ table cell. Let $(EP_k)_r$ mean the $r^{th}$ vector component of the vector $EP_k$, which is defined as a complex-valued continuous function shown as Equation (2).

$$(EP_k)_r = \cos(\omega_r k + \theta_r) + i\sin(\omega_r k + \theta_r) \tag{2}$$

Where $i$ indicates the imaginary part of complex-valued continuous function. And, $\omega_r, r \in [1, d]$ is the period related weights and $\theta_r, r \in [1, d]$ is the phase related vector. $\omega_r$ and $\theta_r$ are learnable parameters. We note a cross-entropy loss function of the positional context embedding as $L_{EP}$.

Thirdly, we represent the lexical (tokenized) contexts of the table layout structures in the table as low dimensional embedding vectors. Let $EL_k$ indicate the embedding vector with dimension $d$ of the lexical context for the $k^{th}$ table cell. The embedding vector $EL_k$ is computed based on the continuous Skip-gram model which is an efficient method for learning high-quality vector representations by using a negative sampling objective. Given the embedding vector $EL_k$ and $EL_i$, the conditional probability of the embedding vector $EL_k$ and $EL_i$ can be expressed by the $Softmax$ function, shown as Equation (3).

$$p(EL_i|EL_k) = \frac{exp((EL_i)^\top EL_k)}{\sum_{j=1}^M exp((EL_j)^\top EL_k)} \tag{3}$$

Where $M$ is the size of vocabulary and $p(EL_i|EL_k)$ means the conditional probability of $EL_k$ and $EL_i$. We note a cross-entropy loss function of the lexical context embedding as $L_{EL}$.

Finally, we represent the colored contexts of the table layout structures in the table as low dimensional embedding vectors. Let $EC_k$ indicate the embedding vector with dimension $d$ of the colored context in the $k^{th}$ table cell. And, let $EC_{k,red}, EC_{k,green}, EC_{k,blue}$ indicate the embedding vector with dimension $d$ of the red, green, and blue context in the $k^{th}$ table cell. We note a cross-entropy loss function of the colored context embedding as $L_{EC}$ and sum $EC_k$ by $EC_{k,red}, EC_{k,green}$, and $EC_{k,blue}$, shown as Equation (4).

$$EC_k = EC_{k,red} + EC_{k,green} + EC_{k,blue} \tag{4}$$

So we get an embedding vector $U_k$ of the $k^{th}$ table cell in the layout structure of the table image, shown as Equation (5).

$$U_k = (ES_k \oplus EP_k \oplus EL_k) \odot EC_k \tag{5}$$

Where $A \oplus B$ is the summary of the vectors $A$ and $B$, and $A \odot B$ is the element-wise multiplication between $A$ and $B$.

And, we can compute overall cross-entropy loss function $L$ by $L_{ES}$, $L_{EP}$, $L_{EL}$, and $L_{EC}$, shown as Equation (6).

$$L = \lambda(L_{ES} + L_{EP} + L_{EL}) + (1 - \lambda)L_{EC} \tag{6}$$

Where $\lambda \in (0, 1)$ is an adjustable hyper-parameter.

### 3.3 SELF-ATTENTION LEARNING

In a table structure sequence such as $(tablecell_1\{context_1\}, boundingbox_1\{x_1, y_1, w_1, h_1\}) \rightarrow (tablecell_2\{context_2\}, boundingbox_2\{x_2, y_2, w_2, h_2\}) \rightarrow (tablecell_3\{context_3\}, boundingbox_3 \{x_3, y_3, w_3, h_3\}) \rightarrow ...... \rightarrow (tablecell_m\{context_m\}, boundingbox_m\{x_m, y_m, w_m, h_m\})$ where the $i^{th}$ context $\{context_i\}$ of the sequence indicates a seven tuple $\{row_i, col_i, order_i, token_i, red_i, green_i, blue_i\}$, $i \in \{1, 2, 3, ......, m\}$. Let $U = (U_1, U_2, U_3, ......, U_m)$ indicate a matrix with dimension $d \times m$. We transform the matrix $U$ with dimension $d \times m$ to a query matrix $Q = (Q_1, ..., Q_m) = W_Q U$ with dimension $d \times m$, a key matrix $K = (K_1, ..., K_m) = W_K U$ with dimension $d \times m$, and a value matrix $V = (V_1, ..., V_m) = W_V U$ with dimension $d \times m$. $W_Q, W_K, W_V$ are the learnable parameters with dimension $d \times d$ by using a self-attention learning mechanism based on the contexts of the table layout structure, shown as Equation (7).

$$A = \sum_{i=1}^{m} w_i V_i \tag{7}$$

Where $A$ is an attention vector with dimension $d$. And $w_i, i \in [1, m]$ is the weight of the $i^{th}$ value vector $V_i, i \in [1, m]$ which is computed via the Softmax function shown as Equation (8).

$$w_i = \frac{\exp(\frac{(Q_i K_i^\top)}{|Q_i||K_i|\sqrt{d}})}{\sum_{j=1}^{m} \exp(\frac{(Q_i K_j^\top)}{|Q_i||K_j|\sqrt{d}})}, i \in [1, m] \tag{8}$$

Where $Q_i K_i^\top$ is the dot product of $Q_i$ and $K_i$. $|Q_i|$ and $|K_i|$ are the length of query vector $Q_i$ and key vector $K_i$ with dimension $d$, respectively. In order to make the training process more stable, we multiply $|Q_i|$ and $|K_i|$ with the scaling factor $\sqrt{d}$ of the original Transformer Vaswani et al. (2017). This makes the attention naturally normalized, and thus can have smoother attention values which are less likely to fall into extremes.

Then, we output the probabilities of next table cell with the bounding box in the table layout structure sequence via the Softmax function shown as Equation (9).

$$p(U_j) = \frac{\exp(U_j^\top A)}{\sum_{n=1}^{N} \exp(U_n^\top A)}, j \in [1, N] \tag{9}$$

Where $j$ means the $j^{th}$ table cell, $N$ means the number of table cells, $U_j$ indicates the embedding vector with dimension $d$ of the $j^{th}$ table cell, and $p(U_j)$ is the probability of the $j^{th}$ table cell. The table cell with the maximum probability is predicted as next table cell with the bounding box in the table layout structure sequence.

## 4 EXPERIMENT

We apply our color-wise context-splitting Transformer-based table recognition method to detect the table structures from the scanned PDF documents and recognize the contents of table cells containing the red stamps on some handwritten invoice amounts.

### 4.1 DATASETS

Unlike other machine learning sub-fields such as computer vision or natural language processing (NLP), there are no standard benchmarks for tabular data Grinsztajn et al. (2022). A large-scale TabRecSet Yang et al. (2023) dataset for end-to-end table recognition in the wild is introduced and compared between the TabRecSet dataset and the existing datasets such as the PubTabNet Zhong et al. (2020) dataset, the TableX Desai et al. (2021) dataset, and the PubTables-1M Smock et al. (2022) dataset. The PubTabNet dataset is a public dataset which contains 568k table images with corresponding structured HTML representation and includes 548,592 training samples and 9,115 validation samples. The FinTabNet dataset is a large dataset containing complex tables from the annual reports of the $S\&P$ 500 companies Zheng et al. (2020). Financial tables often have very different styles with fewer graphical lines and larger gaps within each table and more colour variations. We conduct experiments on the PubTabNet and FinTabNet dataset with generated ground-truth for reproducibility to verify the effectiveness of our proposed color-wise context-splitting Transformer-based method for recognizing the table cells containing the red stamps from scanned PDF documents. In the PubTabNet and FinTabNet dataset, the ground truth data is derived from the presence of nearby vertical graphical lines (as detected by a PDF parser) for each cell Zheng et al. (2020). In addition, we collect some real dataset from our project in order to train our proposed color-wise context-splitting Transformer-based table recognition model and compare the results with eight baselines. Our collected real dataset is divided into two parts. The first part is some scanned PDF documents which are obtained by scanning printed documents. In the PDF documents of the fist part, the tables contain some red stamps on them. The second part is some born-digital PDF documents which are obtained by directly saving from Word documents. In the PDF documents of the second part, the tables don't contain any red stamp on them.

### 4.2 BASELINES

TableMaster Ye et al. (2021) is customized based on MASTER Lu et al. (2021) which a powerful image-to-sequence model originally designed for scene text recognition, and includes one branch to predict the HTML item sequence and the other branch to conduct the box regression. LGPMA Qiao et al. (2021) recognizes complicated table structure with local and global pyramid mask alignment. TableFormer Nassar et al. (2022) takes a table image and creates HTML structure predictions and bounding boxes. PP-StructureV2 Li et al. (2022a) introduces an ultra lightweight layout detector PP-PicoDet to divide document images into predefined areas such as text, title, table, and figure. TabTransformer Huang et al. (2020) is a novel deep tabular data modeling architecture for supervised and semi-supervised learning which is built upon self-attention based Transformers. FT-Transformer Gorishniy et al. (2021) is a revisiting simple adaptation of the Transformer architecture for tabular data. MATE Eisenschlos et al. (2021) is a novel Transformer architecture designed to model the structure of web tables which uses sparse attention in a way to allow heads to efficiently attend to either rows or columns in a table. DONUT Kim et al. (2022) is a novel OCR-free VDU model which stands for document understanding Transformer. We use TableMaster Ye et al. (2021), PP-StructureV2 Li et al. (2022a), TableFormer Nassar et al. (2022), LGPMA Qiao et al. (2021), TabTransformer Huang et al. (2020), FT-Transformer Gorishniy et al. (2021), MATE Eisenschlos et al. (2021), and Donut Kim et al. (2022) as eight baselines.

### 4.3 EVALUATION METRICS

We evaluate our proposed color-wise context-splitting Transformer-based model and eight baselines with the field-level F1 score metric which checks whether the extracted field information is in the ground truth. Even a single character is missed, the score assume the field extraction is failed. Although F1 is simple and easy to understand, there are some limitations Kim et al. (2022). First, it does not take into account partial overlaps. Second, it can not measure the predicted structure

such as groups and nested hierarchy. In order to assess overall accuracy and capture multi-hop cell misalignment, we use a Tree-Edit-Distance-based Similarity (TEDS) metric Zhong et al. (2020) which identifies both table structure recognition and OCR errors. In addition, we also use root mean-squared error (RMSE) for table recognition tasks. The same evaluation metrics are applied to validation sets for early stopping.

## 4.4 TRAINING OBJECTIVE

In table recognition stage where we recognize the table cells containing the red stamps on some handwritten signatures, we use Intersection Over Union (IOU) to measure the overlap between two bounding table cells by measuring the correctness of the prediction with the ground truth in the HTML format which is extracted from table image. During the training processes, we need pay attention to some key points including the input size of table image and the minimum area size. For example, when the maximum number of recognizable tokens in a table structure sequence are set to 500, the table images which involve more than 500 tokens will be ignored when calculating the accuracy and the TEDS. In addition, the training data of the PubTabNet dataset involve some ambiguous annotations about empty table cell. In order to train our proposed color-wise context-splitting Transformer-based model, we use an ADAM optimizer with the initializing learning rate 0.001 and then reduce the learning rate to 0.0001.

## 4.5 RESULTS

### 4.5.1 COLORED CONTEXT SPLITTING

After involving the colored context embedding of the table cell with a red stamp on it, our proposed color-wise context-splitting Transformer-based model can correctly detect and recognize the handwritten signed invoice amount under the red stamp in the invoice amount column of the table in Figure 1. For example, Figure 4 shows a result of colored context splitting from the table cell with a red stamp on it.

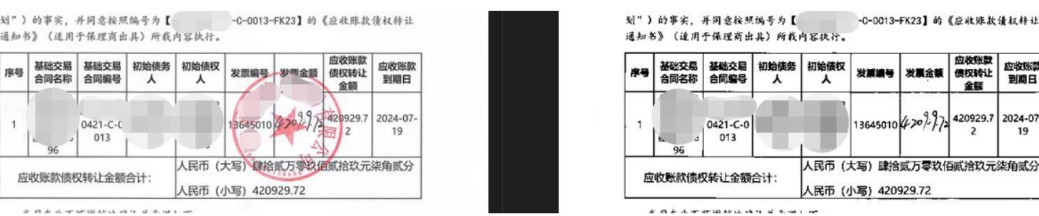

Figure 4: A result of splitting the table cell and the red stamp on it.

### 4.5.2 TABLE RECOGNITION RESULTS ON PUBTABNET AND FINTABNET DATASET

Table 1 shows the $F_1$ (%) and TEDS (%) results of eight baselines and our proposed color-wise context-splitting Transformer-based table recognition method on PubTabNet and FinTabNet dataset.

Table 1: The $F_1$ scores and TEDS results of eight baselines and our proposed color-wise context-splitting Transformer-based table recognition method on PubTabNet and FinTabNet dataset.

| Method | $F_1$ 1 | TEDS 1 | $F_1$ 2 | TEDS 2 |
|---|---|---|---|---|
| TableFormer Nassar et al. (2022) | 77.91 | 96.75 | 77.95 | 96.80 |
| TableMaster Ye et al. (2021) | 77.85 | 96.26 | 77.89 | 96.35 |
| LGPMA Qiao et al. (2021) | 65.74 | 94.70 | 65.78 | 94.77 |
| PPStructure Li et al. (2022a) (+MergeToken) | 76.31 | 95.89 | 76.36 | 95.92 |
| TabTransformer Huang et al. (2020) | 77.61 | 96.68 | 77.68 | 96.75 |
| FT-Transformer Gorishniy et al. (2021) | 76.78 | 95.91 | 76.83 | 95.96 |
| MATE Eisenschlos et al. (2021) | 77.53 | 96.13 | 77.58 | 96.19 |
| DONUT Kim et al. (2022) | 76.91 | 95.98 | 76.99 | 96.06 |
| **Our proposed color-wise Transformer-based method** | **79.95** | **98.81** | **79.99** | **98.89** |

In Table 1, $F_1$ 1 and $F_1$ 2 indicate the $F_1$ scores of the methods on PubTabNet and FinTabNet dataset, respectively. Also, TEDS 1 and TEDS 2 indicate the TEDS results of the methods on PubTabNet and FinTabNet dataset, respectively.

### 4.5.3 TABLE RECOGNITION RESULTS ON SCANNED PDF DOCUMENTS WITH RED STAMPS

For table recognition on scanned PDF documents containing some red stamps on the tables of the PDF documents, the RMSE results of eight baselines and our proposed color-wise context-splitting Transformer-based table recognition method are shown as Table 2.

Table 2: The RMSE results of eight baselines and our proposed color-wise Transformer-based table recognition method on the dataset of scanned PDF documents containing red stamps on the tables.

| Method | RMSE on dataset with red stamps |
|---|---|
| TableFormer Nassar et al. (2022) | 43.9 |
| TableMaster Ye et al. (2021) | 45.3 |
| LGPMA Qiao et al. (2021) | 53.8 |
| PPStructure Li et al. (2022a) | 50.6 |
| TabTransformer Huang et al. (2020) | 39.9 |
| FT-Transformer Gorishniy et al. (2021) | 44.2 |
| MATE Eisenschlos et al. (2021) | 40.1 |
| DONUT Kim et al. (2022) | 43.5 |
| **Our proposed color-wise Transformer-based method** | **31.8** |

In Table 2, the PP-StructureV2 method, the TableMaster method, the TableFormer method, the LGPMA method, the TabTransformer method, the FT-Transformer method, the MATE method, and the DONUT method can't thoroughly distinguish between a handwritten signature and a red stamp very well because it will detect handwritten signature and part of red stamp as the same text box especially when the red stamp happens to be on the handwritten signature.

### 4.6 ABLATION STUDIES

In order to investigate the effectiveness of the spatial context, positional context, lexical context, and colored context, we consider eight different variants in total. Eight variants compared in ablation studies are summarized in Table 3.

Table 3: The RMSE, $F_1$ (%) scores, and TEDS (%) results between eight variants on scanned PDF dataset containing red stamps on the table cells.

| Spatial | Positional | Lexical | Colored | RMSE | $F_1$ | TEDS on dataset with red stamps |
|---|---|---|---|---|---|---|
| ✓ | ✓ | ✓ | ✓ | **31.8** | **80.31** | **98.93** |
| ✓ | ✓ | ✓ | × | 48.9 | 69.88 | 85.96 |
| ✓ | × | ✓ | ✓ | 35.6 | 77.93 | 93.35 |
| × | ✓ | ✓ | ✓ | 41.8 | 73.65 | 89.38 |
| ✓ | ✓ | × | ✓ | 33.6 | 79.12 | 95.39 |
| × | ✓ | ✓ | × | 51.3 | 67.34 | 84.37 |
| ✓ | × | ✓ | × | 53.7 | 65.92 | 83.05 |
| ✓ | ✓ | × | × | 50.1 | 68.23 | 84.55 |
| × | ✓ | × | × | 56.7 | 62.74 | 79.41 |

In Table 3, ✓ indicates that the context embedding is involved and × indicates that the context embedding is NOT involved. When we remove color-wise embedding component from embedding layer, the RMSE, $F_1$ (%) scores, and TEDS (%) results on scanned PDF documents will become poor, shown as the second row in Table 3. In addition, the color-wise embedding component is also applicable for the scenario where the tables contain some stamps of colors other than red. Also, when we remove spatial embedding component from embedding layer, the RMSE, $F_1$ (%) scores, and TEDS (%) results on scanned PDF documents will become poor, shown as the forth row in Table 3. And the accuracy of table recognition from the table containing a red stamp on handwritten invoice amounts will be improved when we replace CSP-PAN with XTab-PAN shown as Figure 3.

## 4.7 LIMITATIONS

In the majority of cases, our proposed color-wise context-splitting Transformer-based table recognition method is effective for recognizing the table cells containing the red stamps on some handwritten invoice amounts. However, it will fail to detect the table structure from a scanned PDF document when this table spans multiple pages of this scanned PDF document. For example, Figure 5 shows that a table contains a red stamp on handwritten invoice amounts and spans multiple pages of a scanned PDF document. Because it fails to detect the layout structures of the table in Figure 5, there are some limitations on our proposed color-wise context-splitting Transformer-based method for recognizing the table cells from a table spanning multiple pages when the table which contains a red stamp on some handwritten invoice amounts spans multiple pages of a scanned PDF document.

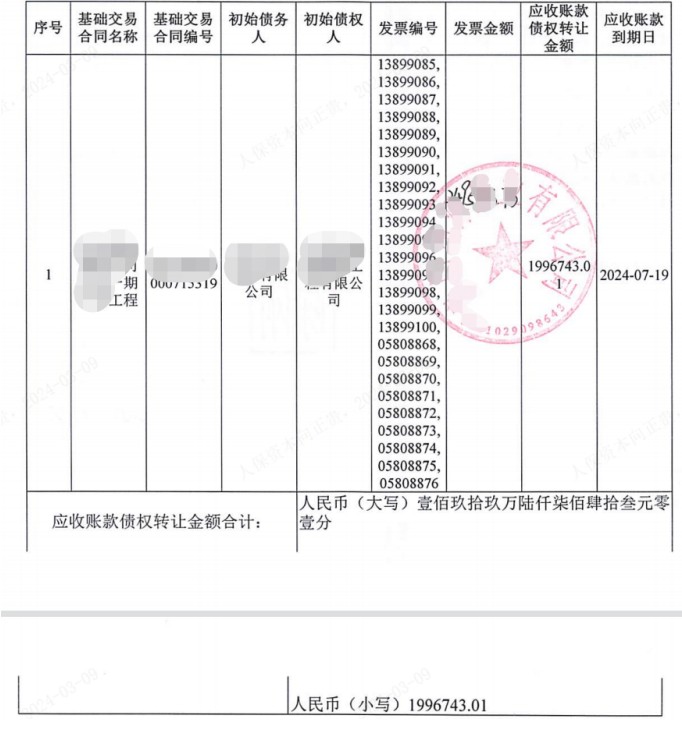

Figure 5: An example on a table containing a red stamp on some handwritten invoice amounts and spanning multiple pages of a scanned PDF document.

## 5 CONCLUSIONS

Tables are a common form in PDF documents. However, it is challenging to recognize the table contents from scanned PDF documents containing some red stamps on some handwritten invoice amounts and reconcile the recognized table contents. In this paper, we propose a color-wise context-splitting Transformer-based table structure analysis and table cell recognition method to recognize the table cells containing red stamps on some handwritten invoice amounts from scanned PDF documents. The contributions of our proposed color-wise context-splitting Transformer-based table recognition method are three-folds. Firstly, we construct efficiently the feature representation of each table cell by fusing spatial, positional, lexical, and colored context features extracted by the context-splitting based decoders. Secondly, we represent the spatial contexts, positional contexts, lexical (tokenized) contexts, and colored contexts of the table cells in a table structure sequence as low dimensional embedding vectors. Finally, we modify the self-attention mechanism of original Transformer Vaswani et al. (2017) by multiplying the length of query vector and the length of key vector with the scaling factor of the original Transformer in order to make the training process more stable. As a result, we improve the recognition accuracy of table cells with a red stamp on handwritten invoice amounts and the efficiency of reconciliation.

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
