# OpenReview forum: "Table Learning Representation from Scanned PDF Documents Containing Some Red Stamps"
_ICLR.cc/2025/Conference — ICLR 2025 Conference Withdrawn Submission_

### Official Review · Reviewer_RRL3 · 2024-11-01

**Soundness:** 1
**Presentation:** 1
**Contribution:** 1
**Rating:** 1
**Confidence:** 3

**Summary:**

The paper addresses a very specific problem in table recognition, the recognition of text that overlaps with red stamps. For that, the authors integrate into the table recognition architecture and embeddding that combines spatial location, position in the table, lexical content and color information. Experiments are conducted on existing table recognition benchmarks and on a specific dataset that includes red stamps.

**Strengths:**

Experiments show that the proposed approach can obtain better results than other SoA methods on standard benchmarks and also on the dataset collected specifically for the problem of recognition with red stamps.

**Weaknesses:**

- The paper addresses a very specific topic. It seems difficult to generalize the proposed approach to other situations.
- It is not clear how the proposed architecture addresses the specific topic of recognition with red stamps. It seems a generic architecture for table recognition and only the introduction of a color embedding seems to have some relation with the problem of red stamps. Nevertheless, this is not well motivated in the paper.
- The description of the proposed architecture is not clear enough. For instance, it is not clear how the embeddings can be computed from the output of the XTab-PAN network (how the cells have been detected to obtain the embedding of each cell?) or how the losses for the embeddings (eq. 6) are defined and computed, or what is exactly the probability obtained as output of the attention module (eq. 9)
- It is not clear the rationale of the experiments with standard benchmarks on table recognition, PubTabNet and FinTabNet, that do not contain specific images with red stamps. In the experiments with the specifically collected dataset, there are no statistics about how many images it contains and how many with red stamps. The description of the metrics used in the experiments should be better explained, why different metrics are used in different benchmarks and what do these metrics exactly evaluate.
- In general, the writing and presentation of the paper should be improved. There are some parts of the description of the method that are difficult to understand. Related work is just a list of mehtods with a brief description without a detailed discussion of advantages and drawbacks of existing methods that could justify the proposed approach.

**Questions:**

See comments above

---

### Official Review · Reviewer_aPf7 · 2024-11-03

**Soundness:** 2
**Presentation:** 2
**Contribution:** 2
**Rating:** 3
**Confidence:** 4

**Summary:**

This paper presents an architecture for table recognition in document images. It is focused in a particular case of tables that have handwritten total amounts with a red stamp on it. A Tansformer based architecture is proposed for the recognition of table cells. The proposed architecture has three components. First, the layout structure of the table is detected, considering that there are some handwritten amounts with red stamps. Second, and embedding representation is proposed for table cells. The embedding combines spatial context, position context, lexical context and colored context embedding. Third a stack of Transformer-based self-attention encoders is used to recognize the cross-modality table cells. The proposed architecture is evaluated with PubTabNet and FinTabNet datasets, and compared with eight SotA methods.

**Strengths:**

Table recognition is a popular topic in Document Intelligence, with high impact in large document processing applications. This paper faces this problem with the added difficulty of recognizing both printed and handwritten cells, and with artefacts, like stamps, difficulting the objective.

The comparative evaluation is quite convincing in terms of the figures that are reported in the tables. However, providing a more comprehensive discussion of the results would have been good.

**Weaknesses:**

The paper is very difficult to follow. There are many technical details, and notations that would require further explanations in order to facilitate the reproducibility. Authors should have used the supplementary pages to provide implementation details. Examples of this observation are: The state of the art section is highly dense. There are big paragraphs with a long enumeration of references, but there is no critical analysis that motivates the contributions of this paper. I recommend to break the paragraphs, grouping the references by methodologies, stating some pros and cons of them. In the methodology section, for example subsection 3.3 (self-attention learning) is very dense too, with big equations. Some sentences giving the intuitive idea, relating them to the data of the table cells would help to understand.

The focus of the paper is misleading. The objective seems to be very specific for a particular set of documents with handwritten total amount cells, having a red stamp. However, algorithms for table recognition are proposed. The evaluation performance is also done using two public datasets: PubTabNet and FinTabNet, that do not contain red stamps on handwritten cells.

PubTabNet and FinTabNet datasets do not originally contain the objects that drive the proposal of solutions. Are you synthetically augmenting these datasets to ensure enough data of cells covered by red stamps? If you are generating synthetic data with red stamps, can you provide examples and the procedure to do so? If it is not applicable, because you only detect stamps in the private collection, please give more details of this dataset, are the stamps always folowing the same pattern? Is the method applicable to other types of stamps/colors?

The actual problem is evaluated in a private collection. The results are shown in table 2 but since this dataset does not seem that will be publicly available, it is difficult to assess the good points of the proposed model (beyond the numerical scores). Is this dataset containing different types of stamps, with different colors?

**Questions:**

Please see the previous sections. Additionally, I am providing below a couple of questions:

Is the proposed model generalizable to other types of artifacts (other types of stamps, signatures or handwritten annotations)? Please can you experimentally disccus this generalization, taking some examples that contain other types of artifacts, and showing if the proposed model scales up to deal with them?

The table layout structure analysis is strongly based in PPStructure Li et al. (2022a). What are the main differences and contributions beyond the SotA, and in particular this cited work?

The following reference is repeated (with different years, 2021 and 2023):

Rujiao Long, Wen Wang, Nan Xue, Feiyu Gao3, Zhibo Yang, Yongpan Wang, and Gui-Song Xia.
Parsing table structures in the wild. In ICCV, 2021.

---

### Official Review · Reviewer_JVJ6 · 2024-11-05

**Soundness:** 1
**Presentation:** 1
**Contribution:** 1
**Rating:** 3
**Confidence:** 4

**Summary:**

The authors introduce a Transformer-based approach for table structure analysis and cell recognition, utilizing color-wise context splitting to detect table cells that contain red stamps on handwritten invoice amounts in scanned PDF documents. This method addresses the specific challenge of recognizing mixed  red stamps  content in real scenes.

**Strengths:**

- The motivation is good.
-The proposed challenge of "recognizing table cells containing red stamps on some handwritten invoice amounts" is interesting.

**Weaknesses:**

- The model description is unclear. For example, how is the context-splitting-based decoder implemented? What are the structures of the spatial context decoder and lexical context decoder? What is the final concatenated feature dimension?

- The experiments is not very sound. Table 2 reports experimental results on the dataset of scanned PDF documents containing red stamps on tables, but this is not clearly explained. Was the model retrained using the collected data, or were the weights from previous models directly used for testing?

**Questions:**

- How should "table cells" be represented?

- What is the difference between XTab and XTab-PAN?

- What does "lexical contexts" mean in line 252 on page 5?

- How many real datasets were collected as described in lines 346–352 on page 7? How many are there in the two parts of the real dataset and the born-digital dataset (without any red stamps) respectively? How are the training and testing datasets divided?

---

### Official Review · Reviewer_aYBW · 2024-11-06

**Soundness:** 3
**Presentation:** 3
**Contribution:** 2
**Rating:** 3
**Confidence:** 5

**Summary:**

The paper presents a method for table recognition from scanned pdf documents where tabular cell data is overlapped with red stamps and handwritten amounts. It presents three contributions: 1. It proposes a method to recognize the layout structure of the table by using XTab-PAN based backbone and 4 context-splitting-based decoders. 2. Using spatial context, position context, lexical and colored context embeddings, it represents the table cells as context-splitting embedding vectors by combining the 4 embeddings. Overall cross-entropy loss is also computed by combining each individual embedding losses. 3. Lastly, it applies a stack of Transformer-based self-attention encoders by multiplying |Q| and |K| with scaling factor sqrt(d) to normalize the attention, and to stabilize training.

The approach is compared with 8 known SOTA techniques, and performs better on RMSE, F1 and TEDS scores on datasets like PubTabNet, FinTabNet and a custom dataset. Ablation studies on 4 context-splitting techniques on decoder underscores their individual and combined importance.

**Strengths:**

The primary contribution and strength of the paper is in its context-splitting architecture, which uses XTab-PAN-based backbone and 4 context-splitting decoders: spatial, positional, lexical and color. While context-splitting decoders, by themselves is not an innovation (used in TabTransformer and MATE), using color-wise context splitting is a good extension of an existing idea to solve overlapping red stamps. The ablation study confirms the importance of color-wise context splitting.

Multiplying the length of query and key vectors with the scaling factor normalizes attention and makes training more stable. While not a true innovation, there is a twist to 'Attention is all you need' paper as the paper proposes to multiply with sqrt (d) instead of 1/sqrt(d).

**Weaknesses:**

1. The primary novelty in the paper is the introduction of color-wise context splitting in the decoder. This is very specific to the red-stamps on tables problem, and seems insufficient to warrant enough novelty in a generalized domain.
2. The chosen SOTA techniques aren't focused on the problem at hand - overlapping red stamps on tabular text.
3. Datasets in section 4.1 do not cover the details of custom vs standard datasets, their individual challenges. In fact, the pdf documents in custom datasets do not contain the red stamps - the primary goal of the paper.
4. No baselines on red stamp over tabular text are provided.
5. Ablation section 4.6 does not cover any ablation studies across sections/innovations 4.1 vs 4.2 vs 4.3.
6. The solution should be generalization enough for any overlapping text (very common in receipts, signatures etc.) rather than being focused on a very specific and peculiar case of red stamps on tabular data. It makes it unclear if the context-splitting a tabular innovation or an innovation to solve red-stamp overlapping issue, or does it extend beyond these?

**Questions:**

1. Why the length of query and key vectors was multiplied by the scale instead of dividing by it as in the original paper? What's the intuition behind it? Does it perform better than original? Can we add an ablation study to do it similar to the original paper?
2. Why is EC or L_EC in equations 5 & 6 treated differently (by doing element-wise multiplication or (1-lambda) addition?
3. Why are red stamps not included in the custom dataset? What are the baselines across the standard and custom datasets, and on those separately?

---

### Official Review · Reviewer_C8jT · 2024-11-09

**Soundness:** 2
**Presentation:** 3
**Contribution:** 2
**Rating:** 3
**Confidence:** 3

**Summary:**

The paper proposes a color-wise context-splitting Transformer-based model for table recognition in scanned PDFs, focusing on handling red stamps on handwritten elements. The model introduces embedding mechanisms (spatial, positional, lexical, and color) and reconstructs the table from different scales.

**Strengths:**

1. The paper addresses a relatively underexplored area: the recognition of handwritten invoices with stamps, contributing a preliminary exploration in this domain.

**Weaknesses:**

1. The focus of the paper is quite narrow, proposing a solution for a specific problem with limited innovation. Abstracting the problem to a broader context might provide futher insights for future work.
2. The Figure 3 is unreadable. While the paper introduces the XTab-PAN architecture, XTab is originally intended for learning transferable knowledge across heterogeneous tables in the table deep learning field. In this paper, however, it appears that XTab-PAN is merely used to handle multi-scale table fusion, lacking a detailed explanation of the design rationale.
3. In the experimental section, the ablation study only explores the encoding strategies and does not provide effective support for the role of XTab-PAN.
4. Additionally, it is evident that color encoding critically influences model performance, as all experiments omitting color encoding lag significantly behind existing models, suggesting limited generalization of the proposed approach. A stamp of a different color might cause the model to fail. It seems that a simple stamp recognizer could provide similar performance improvements across all models.

**Questions:**

Please refer to Weaknesses 2, 3, and 4.

---

### Note · Authors · 2024-11-13

I have read and agree with the venue's withdrawal policy on behalf of myself and my co-authors.